# Ubiquitous Love or Not? Animal Welfare and Animal-Informed Consent in Giant Panda Tourism

**DOI:** 10.3390/ani13040718

**Published:** 2023-02-17

**Authors:** David A. Fennell, Yulei Guo

**Affiliations:** 1Department of Geography & Tourism Studies, Brock University, St. Catharines, ON L2S 3A1, Canada; 2Chengdu Research Base of Giant Panda Breeding Research Department, Chengdu 610081, China

**Keywords:** tourism, giant panda, animal-informed consent, animal welfare

## Abstract

**Simple Summary:**

Understanding the relationship that tourists have with the giant panda can be taken-for-granted given this species’ iconic status and cuteness. Based on a self-reporting questionnaire, we found that Chinese “fans” of giant pandas generally paid greater attention to the welfare of the captive pandas in comparison to “non-fans”. Additionally, fans tend to perceive the welfare of captive pandas more positively, who tend to believe that giant pandas offer more prominent consent to being used as tourist attractions. This study provides new insight into an understanding of the human–panda connection. The ubiquitous love we have for pandas can translate into a better life for these animals down the road.

**Abstract:**

Scholars argue that the ubiquity of the “virtual panda”—the panda people meet in zoos and consume as souvenirs, online memes, or videos—exists in a state of hybridity between wild and domesticated. The species has garnered a significant amount of attention because of their iconic status and because of how cute they are to an adoring crowd. However, given the degree of regard tourists have for the panda, there is a dearth of research on different types of visitors to captive panda venues. In filling this gap, we investigated (1) how deeply Chinese “fans” and “non-fans” consider the welfare of captive giant pandas, and (2) if these groups differ in their assessment of whether giant pandas consent to being used as tourist attractions. In both aims, we apply a recent model on animal welfare and animal consent to giant pandas of the Chengdu Research Base of Giant Panda Breeding.

## 1. Introduction

China’s giant panda has value on many fronts. The species has political value in the form of panda diplomacy, or the sending of pandas to other countries for diplomatic reasons [1], cultural value as a national icon [2,3], conservation value both inside China and globally [4], and touristic value as a revenue generator [5]. In 2022, the number of captive pandas doubled over the course of the previous decade to 673 individuals [6]. The wild panda population increased to 1864 individuals based on an expanding network (parks) of the panda’s natural habitat [7], prompting the IUCN in 2016 to reclassify the giant panda from ”endangered” to ”vulnerable” [8]. The conservation value of pandas is estimated to be worth between US$ 2.6–6.9 billion per year, some 27× more than the cost to the Chinese government to facilitate the panda’s conservation [7]. The giant panda has remained a global celebrity in zoos since the exhibition of the first panda, Su-Lin, outside of China in 1937, renowned as “a sensation” [9]. By 2019, 27 zoos in 21 countries outside of mainland China hosted pandas, with significant economic advantages. Pandas at the National Zoo in Washington D.C., for example, attracted an estimated 12 million people in the 1970s [10], while in Japan, pandas are leased or loaned to other institutions for the purpose of gaining huge profits [11]. In 2019, the Chengdu Research Base of Giant Panda Breeding (entry ticket RMB¥55 ≈ US$7.8) was visited by 9 million visitors [12].

However, in contrast to other Ursidae family members, such as polar bears [13,14,15,16,17,18,19], brown bears [20,21,22], sloth [23,24], and black bears [25,26], that can be viewed in their natural habitats, the giant panda is strictly protected by the Wildlife Conservation Law of China and the newly established Giant Panda National Park system [27]. Restrictions on interacting with pandas in the wild places captive pandas at the center of the panda-visitor industry, despite the push to re-establish panda in protected areas [28,29,30]. This one-dimensional manner by which to view pandas in captive venues has been discussed by Nicholls [31], who argues that wild pandas have become largely intangible because of their persistent invisibility from human society in wild spaces. Instead, Nicholls [31] suggests the ubiquity of the “virtual panda”—the panda people meet in zoos and consume as souvenirs, online memes, or videos—exists in a state of hybridity between wild and domesticated. We find this state worthy of investigation because in virtual form, as in captivity, society has embraced the panda as one of the most beloved animals of all.

While the captive tourism industry sets the standard for the production and consumption of virtual or hybridized pandas, we argue that this hybrid state has not been sufficiently interrogated [32]. While conservation is a central feature in the presentation of pandas to visitors, little is known about the keeping of pandas, as similar or different, than the practices of captive sites that place individual (e.g., tourists) and organizational (e.g., the breeding facilities interests in profit and growth) interests above the interests of animals [33]. It has been argued, consistently, that the giant panda garners significant attention because of how attractive or “cute” the species is to an adoring crowd [32,34,35,36]. However, does being attractive or cute provide justification for life as a commodity? To be produced and consumed for profit and pleasure? To justify conservation efforts? Hence, the purpose of this paper is to investigate what it means to be an animal with hybrid and virtual significance in contemporary Chinese society by (1) assessing how deeply Chinese “fans” and “non-fans” consider the welfare of captive giant pandas, and (2) investigating if panda fans and non-fans differ in their assessment of whether giant pandas *consent* to being used in the manner described above. In both aims, we apply a recent model by Fennell [33] on animal welfare and animal consent in the animal-based tourism domain to giant pandas of the Chengdu Research Base of Giant Panda Breeding.

### 1.1. Literature Review

The review of literature follows two main streams. The first provides an overview of research on animal welfare as a distinct theoretical and applied perspective in a broader realm of animal ethics scholarship. Included in this review is current research on animal-informed consent in tourism. The second stream discusses the literature on the concept of “fandom” and the establishment of core differences between panda fans and non-fans at the Chengdu Research Base of Giant Panda Breeding.

### 1.2. Animal Welfare and Animal-Informed Consent

The literature on the ethical use of animals in tourism has advanced considerably over the course of the last two decades, with a flurry of activity in recent years [37]. Not only is the theoretical nature of animal ethics being explored through a range of different theories [38], but also how these theories apply in certain practices and for certain species. Examples include animal welfare and elephants [39], animal rights in reference to hunting [40], the balancing of costs and benefits (utilitarianism) in marine wildlife tourism [41], Burns et al., [42] on dingoes at Fraser Island, Australia, and ecofeminism and its application to polar bears in northern Canada [18].

Studies have also sought to provide greater scope and understanding on how personhood, agency, and stakeholder apply to the use of animals in tourism. In the case of personhood, Cavalieri [43] argues that some animals, like the great apes, must be granted life, liberty, and the prohibition of torture in living lives of dignity and respect. Agency has been discussed in the context of sled dogs in Finland, who should be given more freedom, control, and influence in the events of their own lives in striving for responsible sled dog practices [44]. Furthermore, animals are now being discussed regularly in tourism and animal ethics studies as stakeholders [38,45]. Compelling in this narrative is work by Kenehan [46] in the business and organizational culture context who argues, following Evan and Freeman [47], that a stakeholder is anyone who is “vital to the survival and success of the firm” (p. 58). While not referring to animals in tourism as stakeholders, Kenehan [46] argues from an organizational context that animals used in experimentation are actors who are essential in bringing products to market.

A novel advancement in theory and practice for the use of animals in tourism surrounds the concept of animal-informed consent [33,48]. This perspective disrupts the traditional order of human–animal relationships by placing value into the cognitive, emotional, biological, and behavioral state of an animal instead of solely on the needs of tourism operators and tourists. Szydlowski [48] discussed the challenges of receiving consent from elephants in Nepal, while Fennell [33] developed a comprehensive framework on levels of assent and dissent for sled dogs based on the Five Domains model of animal welfare by Mellor et al. [49].

Traditional assessments of animal welfare typically focus on the cognitive and physiological well-being of animals which, if all conditions are met, indicate that an animal is faring well. The Five Freedoms model, for example, focuses on the freedom from hunger and thirst; freedom from discomfort; freedom from pain, injury, and disease; freedom to express normal behavior; and freedom from fear and distress [50]. In contrast, Mellor et al., [49] in their Five Domains model, include cognitive and physiological assessments, but also the subjective experiences of animals. Appendix A provide an overview of these different positive (enhancing welfare) and negative (reducing welfare) conditions according to the following categories: (1) Nutritional conditions (food and water intake); (2) physical environment conditions (fresh air, good ventilation, shelter from adverse weather), (3) health conditions (hygiene, disease, wounds, malnutrition, obesity), (4) behavioral interactions (with the environment, with other animals, and with humans), and (5) the combination of these four sets of conditions which provide an overall assessment of the welfare of animals. An added dimension to Mellor et al.’s [49] system is the ability to grade negative affective impacts due to the range of different ways in which humans use animals. Therefore, for example, aversive training of companion animals could have a range of negative affective impacts from “none” to “very severe”.

Fennell [33] adapted the Mellor et al. [49] model for the purpose of establishing an animal-informed consent continuum, ranging from active animal-informed consent (extremely positive and moderately positive affective impact), passive animal-informed consent (negligibly positive and negligibly negative affective impact), and no animal-informed consent (moderately negative and severely negative affective impacts) and how they correspond to the animal welfare indicators discussed above. Fennell [33] also suggested that empirical data could further support the development of animal-informed consent indicators. In the following, we consider the possibility that panda fans and non-fans perform as evaluators of welfare indicators.

### 1.3. Panda Fans and Consent

The WWF logo of the giant panda has been recognized as one of the most valuable trademarks that has ever been designed [5]. The ability to attract public investment is only one facet of the giant panda’s “celebrity status”. Blewitt [51] shows that the giant panda becomes an animal celebrity within a complex of “image, conservation status, mystery, politics, bizarre diet, and global cultural merchandising”. For the Chinese, Songster [2] suggests that the celebrity status of giant pandas is a part of the story of the rise of the People’s Republic of China in 1949. Hence, the giant panda also symbolizes the nation’s modernity. The wild panda and the captive panda, waved into a virtual hybridity, allow the animal to gain a distinct advantage in being an animal celebrity.

The celebrity status of captive giant panda in zoos and enclosures almost makes questions about the animals’ welfare superfluous compared to other species living in zoos or breeding facilities. Although historical records show that NGOs and international organizations have struggled to improve the welfare of captive giant pandas [11], pandas have now been given unparalleled care and attention by most hosting organizations [52]. For example, Wang Wang and Funi in the Adelaide Zoo live in a more than 3000 square meters enclosure with abundant choice, comfort, and stimuli [53]. In the Netherlands, Xing Ya and Wu Wen enjoy a 3400 square meters Chinese-style enclosure built at the cost of seven million euros [54]. Panda enclosures in the Chengdu Research Base of Giant Panda Breeding are named “Villas”, suggestive of a luxury lifestyle befitting animals of such high status and attractiveness. In 2020, Ueno Zoo opened a new enclosure, Panda no Mori, spanning about 6800 square meters to accommodate the panda couple, Ri Ri and Shin Shin [55]. The name Panda no Mori, literarily translated into “panda forest”, is in line with efforts to implement a design ideology consistent with the panda’s natural habitat.

Just like human celebrities who often have an extravagant lifestyle, the offer of spacious and luxurious enclosures to captive pandas is only one way that humans pay tribute to these animal celebrities. More often, in parallel to fields such as sports, music, and films, enthusiastic communities towards celebrities are formed. We suggest that “panda fan” or “panda fandom” be employed as a term to describe this unique and intense relationship between tourists and captive pandas. Reflecting on Sandvoss’ [56] definition that fandom is the “regular, emotionally involved consumption of a given popular narrative or text”, we show that panda fandom can be described as a regular and emotional involvement with virtual captive pandas through consumptive experiences such as visitation, souvenir collection, and social media networking. Panda fandom is therefore said to be integral to panda tourism and sustains the production, reproduction, and consumption [57] of these hybrid beings.

The connection to their object of interest is at the core of the differences between fans and non-fans. Duffett [58] notes that precisely because fans develop strong emotional bonds to their objects and fans use the attachments to create relations with their “heroes and with each other”, fans are more than consumers. The emotional connection between fans and their chosen object allows for the demonstration of deep knowledge and exceptional expertise about the object [59,60,61,62,63,64]. The introduction, sharing, and engagement with the knowledge and expertise further contribute to the social connection and capitalization between fans and the fan communities [65]. For example, Hills [63] shows that the fan community of the television show Doctor Who has grown in expertise and knowledge on fandom-specialized wikis that has fundamentally contested the official showrunners’ knowledge and experiences. In this study, we place this attachment and knowledge about captive pandas as the key to distinguishing panda fans from non-fans. We show that the invested time, knowledge, and energy in the captive pandas have legitimated panda fans to evaluate and make sense of the relationship through self-identify with objects, and to even express the relationship by labeling the idol as a family member or close friend [66,67,68,69].

Fennell [33] explains that humans often fail to capture the expression and significance of animal-expressed indicators even if humans have played numerous roles, such as owner [70,71,72,73], trainer [74,75], experimental subject [76,77], and even colleague [78] of animals. We acknowledge the self-identification process of the panda fans as they experience and build intense attachments with these animals by exploring whether they have elevated welfare concerns and acceptance of animal-informed consent over their non-fan counterparts.

## 2. Methods

### 2.1. Questionnaire Design

The researchers designed a two-part questionnaire to evaluate panda fans’ and non-fans’ engagement with giant panda welfare indicators and informed consent. The first part of the survey determined whether participants were fans or non-fans. Studies addressing differences between fans and non-fans have often categorized the two groups based on participants’ self-evaluation [79,80,81]. For example, in their study of celebrity business fans and non-fans, Teng, Liao, and Wei [82] asked respondents to answer the question, “Are you fans of Ashin? Select Yes or No.”, for the purpose of helping researchers to distinguish the two groups. Researchers have also differentiated fans from non-fans based on the two groups’ engagement with social media [83], with the former group actively engaging and following social media updates.

Working from Sandvoss’ [56] definition of fans, above, we differentiate panda fans from non-fans based on self-identification and engagement with panda-related social media. On the one hand, we asked participants to evaluate the extent to which they identify with panda fans on a 6-point Likert scale (1 = Severely negative; 6 = Extremely positive). Second, we asked participants whether they follow panda-themed social media accounts (Yes/No question). Participants who follow panda-themed social media accounts and identified with panda fans positively (including 4 = Negligibly positive, 5 = Moderately positive, and 6 = Extremely positive) were categorized as panda fans. Furthermore, the questionnaire asked participants to evaluate their relationship with the giant panda with the question, “How do you describe your intimate relationship with the panda?” Available answers include “Family members”, “Friends”, “Idol”, “Research object”, “Visitor attraction”, and “Other”. The second part of the survey was built from Fennell’s [33] table of animal-informed consent of 5 main indicators (Table 1). The fourth indicator, “behavioral interactions”, has four sub-indicators concerning the panda’s interactions with the environment, other animals, keepers, and tourists (Table 1).

Based on an experimental test of the questionnaire with tourists, the research team constructed the indicators of conditions. The rationale behind the conditioned question design is participants’ reflections on the test that they did not pay equal attention to each indicator in their visiting experiences. For example, participants in the test phase reflected that if they did not think about the panda’s mental conditions or physical environment at all, it was impossible to evaluate these two indicators. Given the feedback, the research team designed a conditioned questionnaire with an 8-indicator matrix for participants to evaluate their knowledge or observations of the pandas first. The matrix required participants to evaluate their observations of the pandas through the question, “I pay close attention to the panda’s nutrition/physical environmental conditions/health conditions/interactions with the environment/interactions with other animals/interactions with keepers/interactions with tourists/mental conditions”. The participants could suggest that they were either concerned with one, a few, or all eight indicators (answer “yes”), or they did not pay sufficient attention (answer “I do not know”). For example, participants offering “yes” to the panda’s nutrition were then directed to the evaluation of the panda’s water supply, food quality, food quantity, and food variety. With the answer “I do not know”, participants did not need to evaluate the details for nutrition. Evaluations of the details were all based on the 6-Likert scale (1 = Severely negative; 6 = Extremely positive). Demographic data, including gender, age, occupation, educational background, and place of residence, were collected at the beginning of the questionnaire.

### 2.2. Data Collection

Based on long-term fieldwork experiences at the base, the research team collected the questionnaire at the panda base’s exit and panda nursery house to ensure the data’s quality. The questionnaire was published on Wenjuanxing, a survey data collection website, and issued a QR code, which the research team printed out to allow participants to access it in the field. Wenjuanxing automatically recorded all data and ensured all questionnaires were complete. The research team collected data from 14 November to 20 November 2022, generating a pool of 221 respondents. Four questionnaires had unrealistic age entries and were excluded from the pool, leaving a total of 217 valid samples. On average, a questionnaire took 135.8 s to finish. All participants were Chinese, since the ongoing COVID-19 pandemic still puts international tourists in China on hold. In November, the pandemic and control measures continued to influence the number of visitors at the panda base. All participants contributed to the questionnaire voluntarily with prior consent before they scanned the QR code with their own mobile phones. In exchange, the research team offered panda magazines for the participants’ time. The research team recognizes the use of the QR code as a means to access the questionnaire may pose a barrier for tourists with less internet literacy (e.g., tourists over the age of 45 and with a less-than-college degree). Additionally, using panda magazines as an incentive could lead to further differences (e.g., gender, age, and level of education) in the sample. Table 2 profiles participants based on their demographic characteristics.

### 2.3. Data Preparation

The study identifies two groups, panda fans and non-fans, depending on the participants’ self-identification with fans and consumption of social media content. The two conditions allowed us to identify 117 panda fans (53.9%) and 100 non-fans (46.1%). We conducted *t*-tests to evaluate panda welfare differences between panda fans and non-panda fans (Figure 1).

## 3. Results

### 3.1. Panda Consent Indicators

Figure 2 shows that identified panda fans paid greater attention to all panda consent indicators. In contrast, non-fans were more likely to suggest that they did not know or care about these indicators. This finding demonstrates that panda fans, in general, have invested more time and effort into knowing about pandas. We also note that fans (47.47%) and non-fans (29.03%) tend to carefully investigate the pandas’ physical environmental conditions. Fans (33.64%) addressed the mental conditions of the pandas the least, while non-fans (17.79%) paid the least attention to the panda’s nutrition.

Regarding participants’ relationship with pandas (Figure 2), 195 fans and non-fans answered the question: “how do you describe your intimate relationship with the panda” (22 non-fans skipped this question). Results indicate that most panda fans (47.86%) and non-fans (31.00%) tended to view pandas as their friends. More non-fans (25.00%) than fans (20.51%) considered the panda as mere tourist attractions. Fans (14.53%) also have a stronger tendency to view pandas as family members than non-fans (6.00%). The fans, in general, demonstrated a stronger and more intimate connection with the pandas by including the animal as friends and families.

### 3.2. Panda Welfare and Consent Indicators in Detail

Table 3 and Figure 3 compare the performances of panda fans and non-fans on panda welfare and consent indicators. We note that, on average, all indicators were given more than 5 points (between moderately positive to extremely positive) in this study, showcasing the care and attention that giant pandas receive at the panda base. The results further affirm Li’s [52] observation that giant pandas have been given unprecedented welfare benefits in China since the 1990s. Table 3 affirms that the giant pandas are, at least from the point of view of visitors, experiencing the best institutional facilities and relationship with humans possible. We note that panda fans scaled all indicators on average higher than non-panda fans, except for enclosure enrichments.

The results indicate that fans and non-fans ranked the pandas’ interaction with keepers highest (4.3a and 4.3b). Fans believed that keepers are nice and kind (M = 5.80) and skillful and well-trained (M = 5.81). Non-fans had also rated keepers highly. However, we note that no significant difference can be observed in this indicator between fans and non-fans (*p* > 0.05).

Panda fans rated the enclosures’ enrichment (2b) lowest (M = 5.25) in all fan-generated scales. Although panda fans and non-fans did not demonstrate a significant difference in evaluating the enclosure enrichments (*p* = 0.790), it is the only time that fans generated a lower scale than non-fans (M = 5.30) on all indicators. In addition, in all evaluations non-fans were least positive in evaluating the panda’s consent to being a tourist attraction (5a) (M = 5.05). Additionally, we note that fans (M = 5.27) were more conserved in evaluating 5a.

Panda fans and non-fans only differ significantly (*p* = 0.010) in the possibility that pandas could express to humans their consent to being used as tourist attractions (5b). Fans believed firmly in the pandas’ ability to express their feelings to human beings while being a tourist attraction (M = 5.63), whereas non-fans only confirmed the panda’s ability moderately.

## 4. Discussion

In their evaluation of Zoo Atlanta’s Giant Panda Conservation Center, Wilson et al. [84] show that 23 staff and 145 zoo visitors evaluated the exhibit favorably in terms of its design, construction, enclosure, working experiences, and visitor experiences. On a 5-point scale, the staff offered a mean rating of 3.64 for the panda enclosures and their own working space, while visitors provided an average rating of 4.50 for their visits. The favored evaluation results justify Zoo Atlanta’s goal as the organization strives to provide the best possible environment for the pandas and offer visitors educational opportunities [84]. Barua ([5], p. 9) studied the panda enclosure enrichment projects executed around zoos and organizations to improve the enclosures’ biological relevance. Nicholls [31] explains that the enriched enclosures provided for pandas aim at improving animal welfare on the one hand and increasing the species’ productivity on the other hand. Our study is confirmation of the above research results. Giant pandas at the Panda Base, from the point of view of the tourists, are well taken care of and enjoy a very comfortable lifestyle [52]. All animal welfare indicators fall between moderately positive to extremely positive. We suggest that the current welfare indicators of giant pandas substantiate and justify the possible efforts and investment that the tourism industry could use to endorse animals in this industry.

A critical finding of this study is that panda fans and non-fans were “invested” at different (and significant) levels in pandas. Panda fans evaluated the details of welfare and consent indicators at a higher level than non-fans, who tended to be less concerned about the pandas. However, our study shows that panda fans’ evaluation of the panda’s welfare does not differ from non-fans substantially in all indicators, except for the panda’s expression of consent to being used as a tourist attraction (5b). Panda fans believed that pandas willingly play the role of a tourist attraction and would be able to express this willingness. The study also shows that fans more passionately build intimate relationships with the pandas as they identified the animals as their friends and family members. However, this intimate relationship between panda fans and pandas does not support the hypothesis that panda fans know or understand pandas better than non-fans. Instead, this panda fandom could mean that fans deeply anthropomorphize pandas [85], as fans believe they could experience the panda’s feelings and subjectivity. The tourist–panda relationship, building on the notion of fandom, may be a starting point for constructing more ethical contact between the two species. However, the fan’s love and passion towards the animal celebrity, it seems, can also be blind without proper education programs. More studies need to determine whether the panda fandom could further support animal welfare indicators evaluation, especially when international tourism is allowed in China and at the Panda Base.

Keepers, rather than pandas, received the highest credit from panda fans and non-fans in our study. Keepers are believed to be professionals who are kind and nice to the animals. In contrast, neither fans nor non-fans merited the panda’s subjectivity highly, as both had the least concerns with the panda’s mental conditions. Fennell [33] writes that animals can speak for themselves, but humans have missed these “voices” because of ignorance and self-interest. The results in this study support Fennell’s statement, showing that panda fans might also have, wittingly or unwittingly, prioritized human self-interest over the interests of their fanned animal. The professional care, kindness, and attention giant pandas have received mirror the idealized harmony and companionship between the animal world and human beings, but these human efforts and idealizations can also suppress giant panda voices.

The findings of our study indicate that animal welfare evaluations of the nature conducted at the panda base were a novel experience for the Chinese public. Even though the researchers carefully translated the indicators into lay words that participants could understand, the researchers still received questions from participants that some questions/statements were completely new to these participants. We selected panda fans as the target research group because we believed that fans were more concerned about panda welfare and could reflect more thoughtfully on the indicators. While animal protection and welfare are becoming more important in China, Li [52] suggests that the nation is “almost 200 years behind the most developed countries in terms of animal protection legislation”. As such, we, as researchers, felt a deep need to enhance the public’s literacy on animal welfare. A good starting point for this discussion and action would be participants such as panda fans, who have invested in these animals to a greater extent than non-fans. Whether the role of animal fans can become a springboard for further animal protection and welfare enhancement demands more intensive study.

We further contend that captive animal sites, like the Panda Base, generate passionate, emotional responses to pandas, while the pandas themselves are being used in ways that are mainly focused on pleasure and profit, even though conservation is a desired end. As such, we need to move beyond the self in turning our attention to animals not simply as objects to satisfy our curiosity, or indeed political and economic ends, but rather as subjects that exist independent of these ends—or at least alongside such ends. Time will tell how successful breeding sites will be at repopulating the panda’s wild habitat, and the fate of the panda at such time. Until then, the panda exists in a virtual and hybrid state where the few are required to be “flag bearers” for the success of the species and the amusement of followers.

## 5. Conclusions

The purpose of this paper was to investigate how deeply Chinese “fans” and “non-fans” consider the welfare of captive giant pandas, and to better understand if panda fans and non-fans differ in their assessment of whether giant pandas *consent* to being used as tourist attractions. Our general conclusion is that panda fans paid greater attention to all panda consent indicators over their non-fan counterparts. Our approach, especially the latter question, provides new inroads into the relationship that people have with charismatic animals like pandas. We view our research as a benchmark from which to build more of an understanding of the human–panda connection, and if the ubiquitous love we have for pandas translates into a better life for these animals down the road. Furthermore, we argue that cross-cultural research will provide a better foundation of knowledge in this area (as tourism to China and the Panda base reopens in the future), as well as efforts to refine the constructs investigated in our study.

## Figures and Tables

**Figure 1 animals-13-00718-f001:**
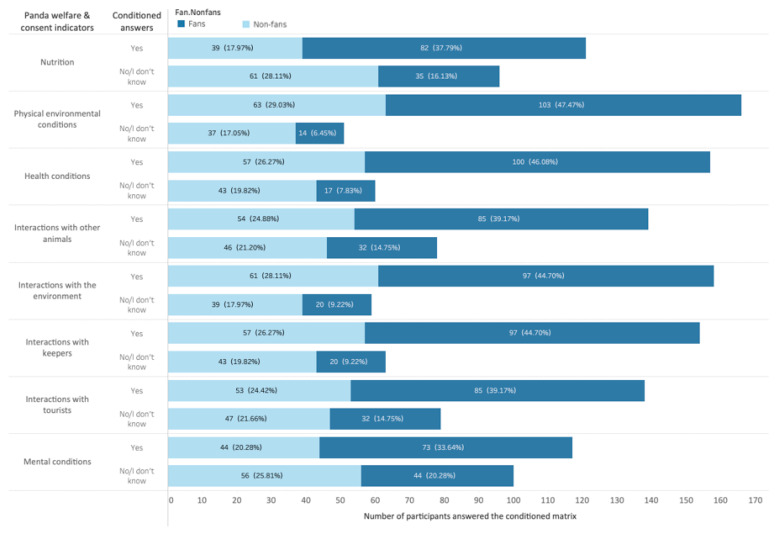
Conditioned answers on eight indicators of fans and non-fans.

**Figure 2 animals-13-00718-f002:**
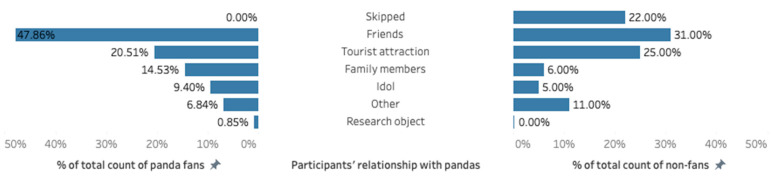
Participants’ relationship with pandas.

**Figure 3 animals-13-00718-f003:**
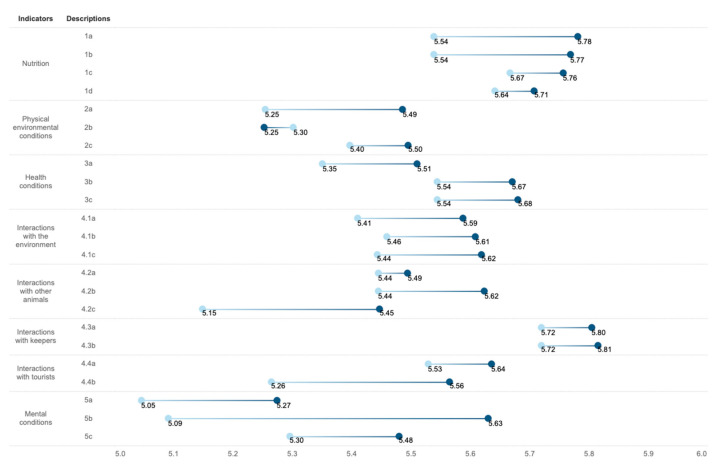
Means of panda welfare and consent indicators by panda fans and non-fans.

**Table 1 animals-13-00718-t001:** Panda welfare and consent indicators.

Indicator		Description		
1. Nutrition		a. Water supply	b. Food quantity	c. Food quality d. Variety
2. Physical environmental conditions		a. Spacious enclosures	b. Enriched enclosures	c. Enclosures protect pandas against thermal extremes
3. Health conditions		a. The panda does not suffer from acute and chronic injuries	b. Healthcare treatment	c. The panda is fit and attends to exercises
4. Behavioral interaction	4.1 Environment	a. Enclosures reflect the panda’s life stage	b. Males and females are separated	c. Multiple choices to move and be active
	4.2 Other animals	a. The panda socializes with a group	b. The panda plays with its peers	c. The panda enjoys community life
	4.3 Keepers	a. Keepers are kind and friendly	b. Keepers are qualified and skillful	
	4.4 Tourists	a. The panda is at ease despite the presence of tourists	b. The panda can respond to my attention	
5. Mental conditions		a. Pandas give consent to being used as tourist attractions.	b. It is possible for pandas to express to humans their consent to being used as tourist attractions	c. Humans can recognize if pandas are expressing their consent to being used as tourist attractions

**Table 2 animals-13-00718-t002:** Demographic summary of participants.

Measure	*N*	*%*
Age	217	Mean = 28.3; SD = 9.10
Gender		
Female	139	64.1%
Male	78	35.9%
Level of education		
Less than college	24	11.1%
Some college	50	23.0%
Undergraduate	97	44.7%
Postgraduate and above	46	21.2%
Occupation		
Full-time	111	51.2%
Student	48	22.1%
Freelance	32	14.7%
Unemployed	8	3.7%
Retired	5	2.3%
Other	13	6.0%
Place of residence		
Chengdu	81	37.3%
First-tier cities	44	20.3%
Other regions in Sichuan	11	5.1%
Provincial capitals	40	18.4%
Other	41	18.9%

**Table 3 animals-13-00718-t003:** *t*-tests on panda welfare and consent indicators and details.

Indicators		Description	Fans			Non-Fans				
			*n*	*M*	SD	*n*	*M*	SD	*t*	*p*
Nutrition		1a. Water supply	82	5.78	0.52	39	5.54	0.82	1.684	0.098
		1b. Food quantity	82	5.71	0.64	39	5.64	0.87	0.472	0.638
		1c. Food quality	82	5.77	0.50	39	5.54	0.88	1.511	0.137
		1d. Food variety	82	5.76	0.60	39	5.67	0.77	0.697	0.487
Physical environmental conditions		2a. Spacious enclosures	103	5.49	0.92	63	5.52	1.05	1.495	0.137
		2b. Enriched enclosures	103	5.25	1.19	63	5.30	1.09	0.266	0.790
		2c. Enclosures protect pandas against thermal extremes	103	5.50	0.88	63	5.40	0.93	0.683	0.496
Health conditions		3a. The panda does not suffer from acute and chronic injuries	100	5.51	0.89	57	5.35	1.20	0.944	0.347
		3b. Healthcare treatment	100	5.67	0.64	57	5.54	0.83	1.070	0.286
		3c. The panda is fit and attends to exercises	100	5.68	0.68	57	5.54	0.87	1.089	0.287
Behavioral interactions	Environment	4.1a. Enclosures reflect the panda’s life stage	97	5.59	0.83	61	5.41	0.99	1.219	0.225
		4.1b. Males and females are separated	97	5.61	0.73	61	5.46	1.01	1.002	0.319
		4.1c. Multiple choices to move and be active	97	5.62	0.77	61	5.44	0.94	1.226	0.223
	Other animals	4.2a. The panda socializes with a group	85	5.49	0.95	54	5.44	1.04	0.290	0.772
		4.2b. The panda plays with its peers	85	5.62	0.72	54	5.44	0.98	1.154	0.252
		4.2c. The panda can enjoy solidarity	85	5.45	0.93	54	5.12	1.30	1.427	0.145
	Keepers	4.3a. Keepers are kind and nice	97	5.80	0.51	57	5.72	0.70	0.862	0.390
		4.3b. Keepers are qualified and skillful	97	5.81	0.53	57	5.72	0.73	0.938	0.350
	Tourists	4.4a. The panda is at ease despite the presence of tourists	85	5.64	0.81	53	5.53	0.85	0.740	0.461
		4.4b. The panda can respond to tourists	85	5.56	0.96	53	5.25	1.11	1.685	0.094
Consent indicators		5a. Pandas give consent to being used as tourist attractions	73	5.27	1.28	44	5.05	1.29	0.930	0.354
		5b. It is possible for pandas to express to humans their consent to being used as tourist attractions	73	5.63	0.72	44	5.09	1.22	2.674	0.010
		8c. It is possible for humans to recognize if pandas are expressing consent	73	5.48	0.96	44	5.30	1.17	0.923	0.358

## Data Availability

Data generating results of the paper are available upon request.

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
