# Peer review of "Ubiquitous Love or Not? Animal Welfare and Animal-Informed Consent in Giant Panda Tourism"

_animals, 2023, doi:10.3390/ani13040718_

Round 1

Reviewer 1 Report

Overall, well-written. I would have liked more guidance in some places.

This paper explores hybridity in contemporary Chinese society using the case of the Giant Panda, as a celebrity status. Specifically, it investigates fandom and perceptions of animal  welfare. Methods include survey data from a panda facility to explore welfare indicators and informed consent – from the view of the tourists. These are grouped as ‘fans’ and ‘non-fans’  and determine capacities in anthropomorphizing panda consent  to justify profit and pleasure. I suggest this paper minor revisions – and an opportunity to distinctly highlight the central argument throughout the paper.

Introduction:

23-27: “In 2022, the number of captive pandas doubled over the course of the previous decade to 673 individuals (Hu & Yu, 2022), with the wild panda population increasing to 1864 individuals based on an expanding network (parks) of the panda’s natural habitat (Wei et al., 2018), prompting the IUCN in 2016 toreclassify the giant panda from "endangered" to "vulnerable" (Swaisgood et al., 2018)”

This sentence is quite long – suggest making it 2 sentences.

45-48: “Restrictions on interacting with pandas in the wild (there is a push to re-establish the connection between the giant panda in wilderness settings, as noted by (Tang et al., 2020; Wei et al., 2015; Yang et al., 2018), places captive pandas at the center of the panda-visitor industry, laying the foundation for panda tourism.”

Rework sentence – hard to digest.

51-53: “Nicholls (2012) suggests the ubiquity of the “virtual 51 panda”—the panda people meet in zoos and consume as souvenirs, online memes, or vid- 52 eos—exists in a state of hybridity between wild and domesticated. “

This seems to be a central tenant – I would leave a transition sentence to tell the reader why this is important. For example, you draw from Nichols to introduce ‘ubiquity’ as a theoretical analytic – also expressed in the title – give me more as to why/how you are using this.

62-63 :” However, does being attractive or cute provide justification for life as a commodity to be produced and consumed for profit and pleasure to justify conservation efforts?“

I suggest – However, does being attractive or cute provide justification for life as a commodity? To be produced and consumed for profit and pleasure? To justify conservation efforts

138 – ‘celebrity status’

Perhaps you could foreground this in the literature review – through hybridity – what are you meaning by ‘celebrities status’ and how has this status shaped the identity of Panda and or Chinese society? This paper would be much stronger if you book marked these things a bit more.

153-156: “The offer of spacious and luxurious enclosures to captive pandas is only one way humans pay tribute to these animal celebrities. In parallel to fields such as sports, music, and films, where enthusiastic communities towards celebrities are formed, we suggest that "panda fan" or “panda fandom” be employed as a term to describe this unique and intense relationship between tourists and captive pandas.

I am confused by this jump – from habitat to fandom.”

316-317: Suggest – Giant pandas at the Panda Base, from the point of view of the tourists, are well taken care of and enjoy a very comfortable lifestyle. All animal welfare indicators fall between moderately positive to extremely positive.

328-336: “The study also shows fans more passionately build intimate relationships with the pandas as they identified the animals as their friends and family members. How-ever, this intimate relationship between panda fans and pandas does not support the hypothesis that panda fans know or understand pandas better than non-fans. Instead, this panda fandom could mean that fans deeply anthropomorphize pandas (Kennedy, 1992) as fans believe they could experience the panda's feelings and subjectivity. More studies need to determine whether the panda fandom could further support animal welfare indicators evaluation, especially when international tourism is allowed in China and at the Panda Base.”

Perhaps a greater discussion on this is necessary – this seems to be a big finding, with some vague and suggestive information. What do you think this might mean? For Chinese Society? For Panda? For hybridity? For wild and domestic panda? Perhaps people assume consent because they anthropomorphize panda, and see panda has ‘happy’ and thus willing to be there? Perhaps they see panda as ‘sad’ because they too feel sad for panda? Your study seems to suggest that because people rate panda’s welfare high within the facilities, then they view panda as providing consent? In either case, has panda chosen or given consent to being there and what might panda choose if given other choices. Maybe further than the purpose of the paper – but greater discussion may be good.

Reviewer 2 Report

General comments: 

I think the idea behind this paper is quite original and creative and I commend the authors for considering fan-based judgments as a method to assessing welfare. I do think that some parts of the paper need clarification and elaboration (see specific comments below):

Specific Comments: 

Line 20: Explain what panda diplomacy means

Lines 27 - 29: This sentence is a little hard to follow. 

Lines 35 - 36: Does this need to be put into quotations or can this be reworded?

Lines 45 - 47: This needs to be clarified/reworded. 

Last paragraph of the Introduction Section: Perhaps elaborate a little more on what contractarianism is for a naive reader. 

Lines 87 - 88: This is a little confusing and needs to be reworded

Lines 93 - 94: Elaborate on what has been discussed more specifically regarding agency and sled dogs in Finland 

Supplementary Tables 1 - 4: In the text of the manuscript on page 3, the authors state that these are aimed at measuring affective states, but not all of the tables really address affective states, or what would be classified as emotions (e.g. hunger is not really an affective state but can lead to affective states). Perhaps this can be explained/clarified?

Lines 172 - 173: States that fans grew in expertise in knowledge but there is nothing stating what type of knowledge or expertise was gained by fans of Doctor Who. 

I do think the methods need to be clarified, especially, the 2nd paragraph on page 5. It almost sounds that the questionnaire was piloted first and then the indicators were constructed, but I am not getting this impression. Also, I find the wording, "conditioned questionnaire design" to be a little confusing. 

Regarding Table 1, you mention 5 main indicators, with one indicator containing subcategories, but the numbering on this table contrasts with this description (which is in the text of the manuscript)

Regarding statistics, were you able to determine whether the differences between fans vs. non-fans on the indicators was significant? I did not see this anywhere. 

Regarding data collection, more females than males were sampled (I see quite a big difference in the percentage. This was not addressed anywhere in the discussion/study limitations. The same goes for education level. 

I think Figure 1 could be clarified/explained better. I found it confusing to follow. 

Line 270: The percentage of 25% for non-fans regarding viewing pandas as family members doesn't seem accurate when referring to Figure 2. 

Lines 314 - 316: You claim that " . . .as noted above, giant pandas in China have received the utmost care possible . . ." but the studies mentioned int the first part of the discussion don't seem to take place in China and the study locations weren't mentioned for some of the studies. This needs clarification. 

Lines 332: I just wanted to mention that you make an excellent point regarding the notion that fans may deeply anthropomorphize pandas. 

Lines 338 - 352: I think you did a great job describing how animal voices have been lost and how this is something to consider. 

Lines 346 - 348: I do think that the information regarding Yaussy's study needs to be clarified. 

On page 12, you mention Author (in press) twice. Please clarify?
